# BrassicaEDB: A Gene Expression Database for Brassica Crops

**DOI:** 10.3390/ijms21165831

**Published:** 2020-08-13

**Authors:** Haoyu Chao, Tian Li, Chaoyu Luo, Hualei Huang, Yingfei Ruan, Xiaodong Li, Yue Niu, Yonghai Fan, Wei Sun, Kai Zhang, Jiana Li, Cunmin Qu, Kun Lu

**Affiliations:** 1College of Agronomy and Biotechnology, Southwest University, Beibei, Chongqing 400715, China; swuthor@163.com (H.C.); chioveelaw@163.com (C.L.); lxd1997xy@163.com (X.L.); ny905319991@163.com (Y.N.); fanyonghai1994@163.com (Y.F.); reginasw@163.com (W.S.); zhangkai2010s@163.com (K.Z.); ljn1950@swu.edu.cn (J.L.); lion4302@163.com (C.Q.); 2Institute of Innovation & Entrepreneurship, Southwest University, Beibei, Chongqing 400715, China; 3State Key Laboratory of Silkworm Genome Biology, Southwest University, Chongqing 400715, China; lit@swu.edu.cn (T.L.); yingfeiruan@126.com (Y.R.); 4Chongqing Key Laboratory of Microsporidia Infection and Control, Southwest University, Chongqing 400715, China; 5Institute of Characteristic Crop Research, Chongqing Academy of Agricultural Sciences, Chongqing 402160, China; hualei_huang@163.com; 6Academy of Agricultural Sciences, Southwest University, Beibei, Chongqing 400715, China

**Keywords:** BrassicaEDB, *Brassica napus*, gene expression profile, RNA-Seq

## Abstract

The genus *Brassica* contains several economically important crops, including rapeseed (*Brassica napus*, 2*n* = 38, AACC), the second largest source of seed oil and protein meal worldwide. However, research in rapeseed is hampered because it is complicated and time-consuming for researchers to access different types of expression data. We therefore developed the *Brassica* Expression Database (BrassicaEDB) for the research community. In the current BrassicaEDB, we only focused on the transcriptome level in rapeseed. We conducted RNA sequencing (RNA-Seq) of 103 tissues from rapeseed cultivar ZhongShuang11 (ZS11) at seven developmental stages (seed germination, seedling, bolting, initial flowering, full-bloom, podding, and maturation). We determined the expression patterns of 101,040 genes via FPKM analysis and displayed the results using the eFP browser. We also analyzed transcriptome data for rapeseed from 70 BioProjects in the SRA database and obtained three types of expression level data (FPKM, TPM, and read counts). We used this information to develop the BrassicaEDB, including “eFP”, “Treatment”, “Coexpression”, and “SRA Project” modules based on gene expression profiles and “Gene Feature”, “qPCR Primer”, and “BLAST” modules based on gene sequences. The BrassicaEDB provides comprehensive gene expression profile information and a user-friendly visualization interface for rapeseed researchers. Using this database, researchers can quickly retrieve the expression level data for target genes in different tissues and in response to different treatments to elucidate gene functions and explore the biology of rapeseed at the transcriptome level.

## 1. Introduction

With the availability and low cost of next-generation sequencing (NGS) technologies, genome-wide omics datasets are currently being generated at higher frequencies than ever before. These resources have rapidly expanded our knowledge of functional genomics in both animals and plants [1]. However, the storage, analysis, management, and maintenance of the massive quantities of data produced by NGS remain quite challenging. In response to accumulating NGS-generated data, various bioinformatics databases have become popular, such as the Gene Expression Omnibus (GEO) from NCBI and ArrayExpress from EBI [2,3]. In previous studies, a range of databases relating to *Brassica* genetics, genomics and related activities have been proposed [4]. The most widely used reference genome for *B. napus* is currently stored in the Genoscope (https://wwwdev.genoscope.cns.fr/brassicanapus/) [5]. However, as a result of structural variations, a single reference genome is unable to cover the entire gene content of a species. Therefore, pangenomics analysis was proposed to ensure genomic diversity within a species is fully represented. The *Brassica napus* pan-genome information resource (BnPIR, http://cbi.hzau.edu.cn/bnapus/) provided eight high-quality reference genomes representing different ecotypes to help researchers get a better understanding of the genome structure and genetic basis of morphotype differentiation in rapeseed [6]. Besides, other *Brassica* pangenome databases were also hosted in the *Brassica* Genome (http://www.brassicagenome.net/) [4]. These databases provide us with high-quality reference genomes and pangenomes, which greatly enable us to identify gene function with more accuracy and convenience.

Among the multiple plant omics datasets, transcriptomic data provide important clues to help predict gene function or reveal hidden molecular mechanisms based on gene expression profiles [7]. Large-scale transcriptome analyses in plants have also led to the development of databases such as PlantExpress [8]. At the transcriptome level, RNA sequencing (RNA-Seq) has emerged as an important approach for comprehensive gene expression analysis. RNA-Seq has several advantages over other techniques [9]. RNA-Seq can be used to comprehensively measure the expression levels of all transcripts in a plant tissue without the need to design probes. Since the emergence of RNA-Seq, RNA-Seq data are continuously being deposited into public databases, such as the NCBI Sequence Read Archive (SRA) database [10]. Other important databases also housing large-scale RNA-Seq data from the animal and plants fields include Silk DB, Melonet-DB, and ePlant [11,12,13]. However, a gene expression database for rapeseed based on RNA-Seq is still lacking.

Rapeseed (*Brassica napus*, 2*n* = 38, AACC) is an allotetraploid species that was formed as a result of spontaneous interspecific hybridization between *Brassica oleracea* (2*n* = 18) and *Brassica rapa* (2*n* = 20) [14]. Since genomic information for rapeseed first became publicly available [5], numerous transcriptome studies have been conducted to enhance our understanding of gene function in this important crop [15,16,17]. However, these gene expression datasets remain to be further integrated and explored.

Here, we constructed the online database *Brassica* Expression Database (BrassicaEDB, https://biodb.swu.edu.cn/brassica/). We generated a large-scale gene expression profile of rapeseed based on RNA-Seq data obtained from 103 tissues from rapeseed cultivar ZS11 during seven developmental stages (germination, seedling, bolting, initial flowering, full-blooming, podding, and maturation) (Figure 1). We chose this elite cultivar for its ultra-high oil content, high lodging resistance, high disease resistance, and low erucic acid and glucosinolate levels, as well as its broad eco-physiological adaptation to different climatic conditions worldwide [18]. Since the Arabidopsis eFP browser has been the first web tool that enables in silico gene expression analysis [19], a number of transcriptome datasets have been integrated in the eFP browser, including different plant tissues and organs in normal conditions and in response to abiotic or biotic stress conditions. We therefore utilized the eFP browser on our website, allowing users to comprehensively view gene expression levels during tissues at various stages of development. To ensure that transcriptome data stored in the SRA database could be further mined and integrated, we also analyzed the transcriptome data from 70 BioProjects, which were obtained from 837 samples related to rapeseed in the SRA database (Figure 1). Three types of gene expression values (FPKM, TPM, and read counts) are provided in the BrassicaEDB. Finally, we developed the “eFP”, “Treatment”, “Coexpression”, and “SRA Project” modules based on gene expression profiles and the “Gene Feature”, “qPCR Primer”, and “BLAST” modules based on gene sequences, providing powerful tools for comprehensive gene expression analysis in rapeseed [20,21].

## 2. Results

### 2.1. RNA-Seq-Based Global Expression Data from 206 Rapeseed Samples

To obtain global expression data for rapeseed, we collected 206 tissue samples from rapeseed cultivar ZS11, including seedlings grown in the greenhouse and plants grown in the field from late autumn to early summer. For each tissue, two biological replicates, each obtained from three independent plants, were collected for RNA-Seq (Appendix A). We presented a cartoon to illustrate 103 rapeseed ZS11 plant tissues in Figure 2, and the detailed information for each tissue was descried in Materials and Methods.

Using the Illumina HiSeq 2500 platform (Illumina Inc., San Diego, CA, USA), we obtained 125 bp of paired-end read data. Using rapeseed genome v.4.1 as a reference, 101,040 genes were used to calculate the fragments per kilobase million (FPKM) in each sample. Then principal component analysis (PCA) and correlation matrix heatmap were performed based on the 206 samples. A total of 91,684 genes (the sum of FPKM in all samples was >10 for each gene) were standardized using Log_2_(FPKM+1), and subjected to R ggfortify package and R corrplot package. An obvious clustering was found in PCA result using the two biological replicates and different clustering was also observed between the 22 tissue groups, suggesting the differences of gene expression profiles among different tissues (Appendix A). Furthermore, the correlation matrix heatmap based on the 206 samples revealed a highly positive correlation between same organ groups (Appendix A), indicating the gene expression profiles were trustworthy. Then the average expression values from two biological replicates were summarized in a gene expression matrix 103 × 101,040 in size. In the 103 samples, 78,224 expressed genes were transcribed (the sum of FPKM in 103 samples was >1 for each gene) (Figure 3A). We calculated the number of expressed genes in the A and C subgenomes separately (the sum of FPKM in each tissue group was > 1 for each gene) and calculated the number of preferentially expressed genes using the R DESeq2 package (adjusted *p*-value <  0.05 and log_2_|fold change| > 1) (Figure 3B). These results suggested that most genes are involved in the regulation of plant growth and development, and a few genes are transcribed at specific tissues during specifically developmental stages.

### 2.2. RNA-Seq-Based Global Expression Data of 837 Samples from SRA Database

To provide users with multiple data types, we downloaded raw transcriptome data for 70 BioProjects from the NCBI SRA database; analyzed the expression patterns of 101,040 genes in 837 samples by RNA-Seq; and determined the FPKM, Transcripts Per Million (TPM), and read counts values of each gene. In total, we generated three 837 × 101,040 gene expression profiles. We also annotated 837 samples with BioProject IDs and run IDs from the NCBI SRA database and provided descriptions of the project, cultivar, sampling stage, sampling tissue, condition, treatment type, treatment reagent, treatment time, PubMed ID, and other important information (Appendix A).

### 2.3. System Architecture and User Interface

The BrassicaEDB was implemented using PostgreSQL (https://www.postgresql.org/), PHP (https://www.php.net/), Apache (http://www.apache.org), Perl (https://www.perl.org/), and React (https://zh-hans.reactjs.org/) on the Linux CentOS 7 (https://www.centos.org) operating system. Our database was supported by the Chado database schema [22]. The dynamic web interface was written in HTML, CSS, and JavaScript. After collecting and analyzing the data, the gene expression profiles and genomic information were loaded into the PostgreSQL database and presented in a user interface framework (Figure 4).

The BrassicaEDB contains three major elements: a search panel on the left that supports dynamic data input, a gene panel that keeps track of the user’s search, and a functional modules panel on the right that displays information based on gene expression profiles and sequences in rapeseed (Figure 5). If a user has a specific search object, they can enter the gene ID in the gene search box to search for the information. If a user only has sequence information, they can utilize the BLAST function to find the genes in the rapeseed sequence database that are similar to the input sequences. When the user enters another query, the previous search results will be recorded as a list at the bottom of the gene search box. Once the gene ID is clicked, a data loading management script sends queries to the database for retrieving and displaying information for each functional module. The user can then explore the following modules in the panel on the right.

#### 2.3.1. Gene Feature Module

The “Gene Feature” module provides basic information about multiple aspects of the selected gene, such as its gene ID, location on the chromosome, and gene structure, including introns, untranslated regions (UTRs), and exons (Figure 6A). Gene Ontology (GO) annotations are available at the bottom. This module also provides DNA, mRNA, coding (CDS), and protein sequences.

#### 2.3.2. eFP Module

The “eFP Map” module displays the expression pattern of the selected gene by dynamically coloring the tissues of a pictographic representation of a plant based on the FPKM from the 103 samples (Figure 6B). The figure is drawn as a vector image, allowing users to obtain good clarity when browsing and to modify the figure easily. Users can select the tissues of interest, explore the expression values of selected genes, choose the color system to change the display style (including absolute and relative mode), and save the figure in SVG format by clicking the “Download SVG” icon. Relative mode allows users to customize the range of gene expression levels, allowing them to easily compare the expression levels of multiple genes within a fixed range. 

The “eFP Chart” mode presents gene expression levels as a histogram sorted from lowest to highest. Users can hone in on samples with high expression levels accurately and rapidly (Appendix A). The “eFP Table” mode includes detailed information and expression values of selected genes in the 103 samples (Appendix A). Users can download sample information and gene expression data by clicking the “Export To Excel” button.

#### 2.3.3. Coexpression Module

Creating gene coexpression networks is a powerful approach for clustering coexpressed genes that are most likely functionally related, speculating about the functions of uncharacterized genes, and detecting genes with similar expression patterns across large amounts of transcriptome data [23]. The “Coexpression” module provides information about gene coexpression relationships based on transcriptome data from the 103 samples (Figure 6C). We used Weighted Gene Coexpression Network Analysis (WGCNA) to construct gene coexpression networks [24]. For this purpose, we selected the top 50% of genes with the highest variance to calculate the weight value for each gene pair. Based on the gene expression profiles of the 103 samples, the top 100 gene pairs with the highest weight values were identified by Pearson correlation coefficient (PCC) analysis. Gene pairs with PCC > 0 and *p*-value < 0.01 were retained. Weight values and PCCs between gene pairs are available for users to view in the “Coexpression” module. This module also presents a visual interface in which the selected gene is depicted as a big ball in the center surrounded by small balls depicting coexpressed genes, each connected to another ball representing the gene with the highest weight value. The “Coexpression” module displays all gene pairs for the selected gene by default. Users can limit the number of gene pairs displayed by inputting number in the box. If the user adjusts the parameters appropriately, the data can be downloaded locally by clicking the “Export To Excel” button.

#### 2.3.4. Treatment Module

The “Treatment” module describes the results of gene expression analysis in rapeseed in response to three types of treatments, including abiotic stress, biotic stress, and chemical treatments. Each project contains all of the gene expression data. There are two modes to view: Table and Chart mode. Table mode presents all information and gene expression data for each sample (Appendix A). Users can download this information by clicking the “Export To Excel” button. Chart mode shows gene expression levels in the form of multiple bar charts (Figure 6D). Users can choose the treatment of interest and explore the expression level of the selected gene under multiple treatments. Gene expression profiles from a selected project can be downloaded for transcriptome analysis in the “Downloads” module (Figure 6H).

#### 2.3.5. SRA Project Module

The “SRA Project” module shows many types of BioProjects to help the user quickly focus on experiments of interest (Figure 6E). In this function, we classified 837 samples into five biological models: biotic stress, abiotic stress, developmental, chemical, and genetic. Users can select a biological experiment and explore the expression values (FPKM, TPM, and read counts) of the selected gene in each sample. All gene expression values can be downloaded locally for further analysis. In addition, descriptions of the experiment, cultivar used in the experiment, sampling stage, sampling tissue, condition, treatment type, treatment reagent, treatment time, PubMed ID, BioProject ID, and run ID in the NCBI SRA database are available by clicking the “Export To Excel” button, allowing users to obtain additional information to help them complete their research.

#### 2.3.6. BLAST and qPCR Primer Modules

The “BLAST” module allows users to compare nucleotide or protein sequences with the *B. napus* database of sequences and identify library sequences that resemble the query sequence (Figure 6F). The BLAST database was built with NCBI BLAST+ 2.10.1 [20]. The “qPCR Primer” module allows users to rapidly select the best primer pair for a gene (Figure 6G). All resources are from qPrimerDB [21].

#### 2.3.7. Other Modules

The “Browse Genes” module shows all available information about a rapeseed gene, including gene ID, the length of the gene and its location on one of the 19 rapeseed chromosomes, and gene note information. Users can click on the gene ID of interest to obtain more information (Appendix A). The “Downloads” module provides all gene expression profiles from the “SRA Project” (Figure 6H), which users can download for transcriptome analysis. The “Links” module contains a collection of other useful databases and websites, which users can browse to obtain more information conveniently. Finally, the “Documents” module generates a user manual that will allow users to easily understand the features of our database and how to obtain information about a gene of interest easily and efficiently.

## 3. Discussion

At present, most rapeseed genomes and transcriptomes data are stored in numerous databases [4,6]. However, the specific database to integrate gene expression data for *Brassica* species, especially for rapeseed, is still absent. Although BnPIR has set up an online web interface to query and visualize gene expression levels, only 40 tissues during flowering stages were included [6]. In this study, we were committed to building a comprehensive gene expression database for rapeseed. We first collected 103 plant tissue materials, covering almost all the tissues in the rapeseed life cycle, and then downloaded 837 public samples from SRA database for RNA-Seq. Overall, The BrassicaEDB provides rapeseed researchers with comprehensive gene expression profiles and a visual interface, filling a gap in the tools available for exploring gene expression in rapeseed and laying the foundation for obtaining a preliminary understanding of gene function in rapeseed at the transcriptome level.

In the future, we plan to improve several aspects of the BrassicaEDB. First, we plan to add data from additional species to the BrassicaEDB. The genus *Brassica* includes several economically important plants. Six species *(Brassica carinata*, *Brassica juncea*, *Brassica napus*, *Brassica oleracea*, *Brassica nigra*, and *Brassica rapa*) evolved by combining chromosomes from three earlier species, as described by the “triangle of U” theory [25]. Thus, in addition to rapeseed, we will include the gene expression profiles of these five other species. Second, we will expand the expression data and RNA types. The transcriptome data obtained from the public databases can be used for analysis at the gene expression level. Using “Brassica” as the query, 8844 transcriptome samples could be retrieved from the SRA database as of 1 July 2020. Clearly, the bulk of transcriptome data remains to be analyzed and further explored. The availability of big data could provide more comprehensive, accurate information about gene function in the future [26]. Finally, we plan to expand the analytical tools available in the BrassicaEDB. To help researchers complete their work more efficiently and conveniently, more tools based on gene expression profiles will be provided in the next version of BrassicaEDB. 

## 4. Materials and Methods

### 4.1. Plant Materials and Growth Conditions

The elite rapeseed cultivar ZS11, which is widely cultivated in China, was selected for developmental transcriptome sequencing. Seeds were germinated in plant growth chambers (PGC Flex; Conviron, Winnipeg, MB, Canada) with a photoperiod of 16 h at 25/18 °C day/night, 60% humidity, and a light level of 250 µmoles/m^2^/s. After germination, the plants were transplanted to the experimental field of Beibei, Chongqing (29°45′ N, 106°22′ E, 238.57 m, CQ) under natural environments. Each plot contained three rows, with 10 plants per row, 20 cm between plants within each row and 30 cm between rows. During the plant lifecycle, 103 different tissues were collected. These tissues included seedling roots (sRo), hypocotyls (Hy), cotyledons (Co) (24, 48, and 72 h after germination; HAG), and germinating seeds (GS) (12 and 24 HAG); roots (Ro) and mature leaves (ML) at the seedling stage; Ro, stems (St), young leaves (YL), ML, buds (Bu), and inflorescence tips (IT) at the bolting stage; Ro, St, YL, ML, pedicels (Ped), IT, sepals (Sep), petals (Pe), carpels (Ca), stamens (Sta), anthers (An), and filaments (Fi) at the initial bloom and full-bloom stages; ML and YL at 10, 24, and 30 days after flowering (DAF); seeds (Se) and silique pericarps (SP) at 15 and 12 regular intervals between 3 and 46 DAF; embryos (Em) and seed coats (SC) at ten stages of seed development (19 to 49 DAF); inner integuments (InI) at 21 and 24 DAF; and outer integuments (OuI) at 24 and 30 DAF. For each sample, two biological replicates, each obtained from three independent plants, were collected and frozen in liquid nitrogen for RNA-Seq.

### 4.2. RNA Isolation and Transcriptome Sequencing

Total RNA was extracted from all tissues using an RNAprep Pure Kit for Plant (Tiangen Biotech, Beijing, China) according to the manufacturer’s instructions and stored at −80 °C until use. Two-hundred-and-six libraries were constructed using a TruSeq RNA Library Prep Kit v2 following standard operating procedures (Illumina, https://www.illumina.com/). All samples were multiplexed per lane of a flow cell, and 125 bp paired-end reads were generated using an Illumina HiSeq 2500 sequencer (Illumina Inc., San Diego, CA, USA).

### 4.3. Public Data Sources

We downloaded the raw sequencing data for 837 samples in 70 BioProjects from the SRA database using fastq-dump from the SRA Toolkit (https://ftp-trace.ncbi.nlm.nih.gov/sra/sdk/2.10.7/), including 169 abiotic stress samples, 211 biotic stress samples, 110 chemical stress samples, 200 developmental samples, and 147 genetic samples, for a total of 2.4 TB of data.

### 4.4. RNA-Seq Data Analysis

The quality of the RNA sequencing reads was examined using FastQC (v0.11.3) (https://www.bioinformatics.babraham.ac.uk/projects/fastqc/). Barcode adaptors from the RNA sequence reads were clipped and low-quality reads removed (read quality < 80 for paired-end reads, read quality < 20 for single end reads) using TRIMMOMATIC software (v0.38) [27]. RNA sequence reads passing the quality filter were aligned to the rapeseed reference genome v.4.1 [4], and rapeseed reference annotation v.5.0 was used as a guide for BAM files using STAR software [4,28]. Quantification of the expression levels of 101,040 genes in each public sample was performed, generating FPKM, TPM, and Read counts values. Cufflinks were used to generate normalized counts in FPKM [29]. Reads or fragments were counted from BAM files using featureCounts, and exons were defined as features at the gene level [30]. The TPM value for each sample was obtained using salmon (v1.0.0) with the parameter “validateMapping” to ensure that all genes would be preserved [31], without using “decoys” parameter in building the mapping-based index progress.

### 4.5. System Architecture and Software for Database Construction

The BrassicaEDB was built with softwares of PostgreSQL 9.6 (http://www.postgresql.org), PHP 7.1 (http://www.php.net), Apache 2.4 (http://www.apache.org) and Perl 5.16 (https://www.perl.org), and all procedures were running on a Linux CentOS 7 (https://www.centos.org) operation system. The Chado database schema was implemented for storing and managing genomic and transcriptomic data [22]. To deal with communications between the website and Chado, the application programming interfaces (API) were developed with Perl and PHP. To provide friendly interfaces for accessing the data, we built the website with JavaScript libraries of React (https://reactjs.org), which supplies many packages and plugins for rendering data and making user interfaces. The BLAST tool was built with NCBI BLAST+ 2.10.1, and it supports searching against selectable and multiple databases [20].

## Figures and Tables

**Figure 1 ijms-21-05831-f001:**
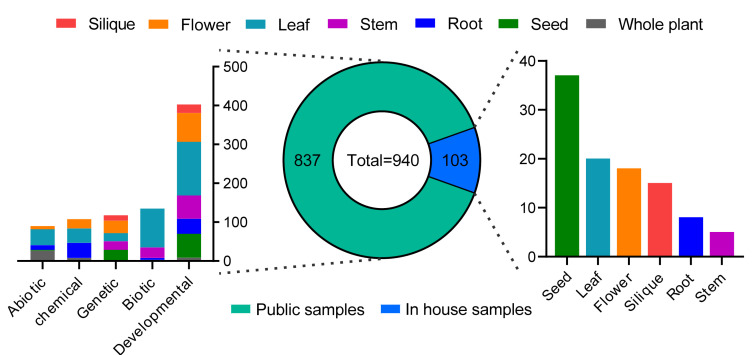
Statistics of samples used in the BrassicaEDB. A total of 940 samples were used to construct the BrassicaEDB, including 103 in house samples and 837 public samples.

**Figure 2 ijms-21-05831-f002:**
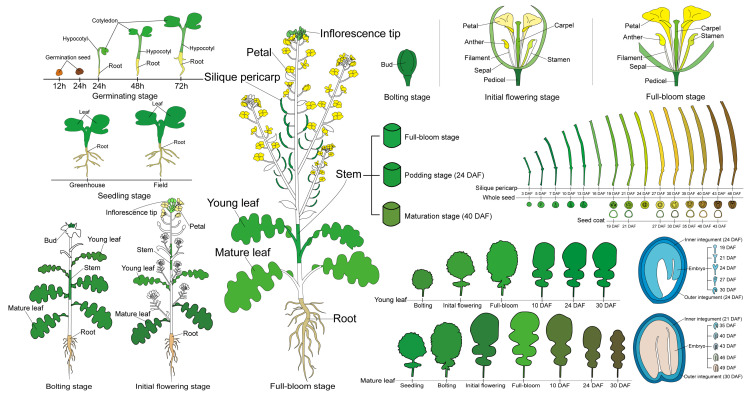
A cartoon illustrating 103 rapeseed ZS11 plant tissues used for RNA-Seq analysis. These tissues included seedling roots (sRo), hypocotyls (Hy), cotyledons (Co) (24, 48, and 72 h after germination; HAG), and germinating seeds (GS) (12 and 24 HAG); roots (Ro) and mature leaves (ML) at the seedling stage; Ro, stems (St), young leaves (YL), ML, buds (Bu), and inflorescence tips (IT) at the bolting stage; Ro, St, YL, ML, pedicels (Ped), IT, sepals (Sep), petals (Pe), carpels (Ca), stamens (Sta), anthers (An), and filaments (Fi) at the initial bloom and full-bloom stages; ML and YL at 10, 24, and 30 days after flowering (DAF); seeds (Se) and silique pericarps (SP) at 15 and 12 regular intervals between 3 and 46 DAF; embryos (Em) and seed coats (SC) at ten stages of seed development (19 to 49 DAF); inner integuments (InI) at 21 and 24 DAF; and outer integuments (OuI) at 24 and 30 DAF.

**Figure 3 ijms-21-05831-f003:**
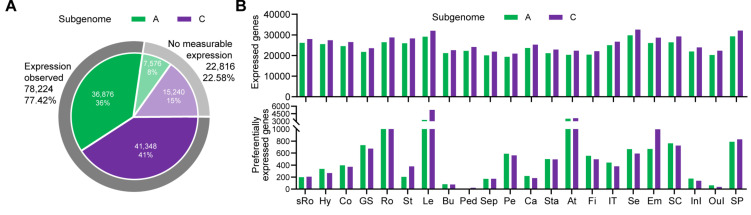
Statistics of expressed genes in A and C subgenomes in rapeseed. (**A**) Gene expression number in the A and C subgenomes. The number of genes with expression observed was 78,224, in which the A subgenome included 36,876 genes (36%) and the C subgenome included 41,348 genes (41%). The number of genes with no measurable expression was 22,816, in which the A sub-genome included 7576 genes (8%) and the C subgenome included 15,240 genes (15%). (**B**) Gene expression and preferentially expression number. The upper bar chart shows the number of expressed genes in 22 tissue groups. The lower bar chart shows the number of preferentially expressed genes in 22 tissue groups. Green represents the A subgenome and purple represents the C subgenome.

**Figure 4 ijms-21-05831-f004:**
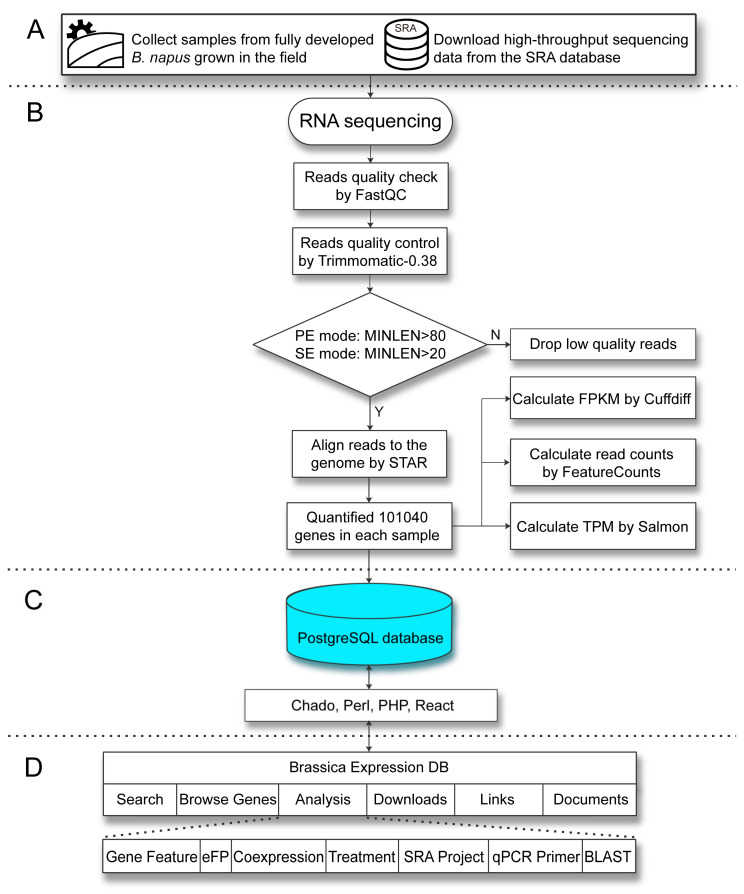
Workflow for the development of the BrassicaEDB. (**A**) Data sources of BrassicaEDB. (**B**) Workflow for the RNA-Seq. Gene expression profils are loaded into the PostgreSQL database. (**C**) Implementation of BrassicaEDB via the integration of different programs. (**D**) Organization of BrassicaEDB.

**Figure 5 ijms-21-05831-f005:**
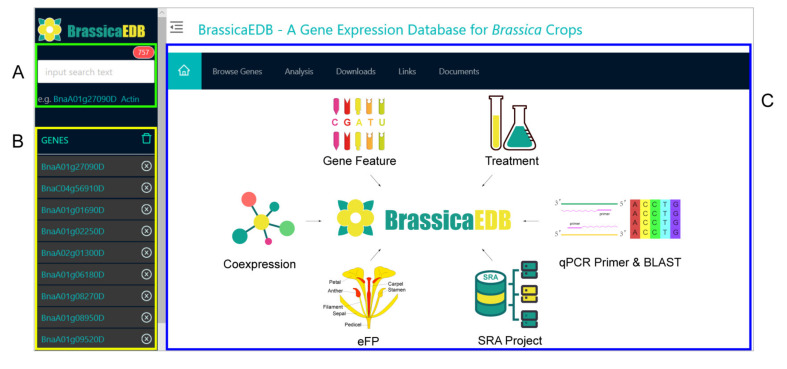
The hompage of the BrassicaEDB. (**A**) search panel, (**B**) gene panel, (**C**) functional modules panel.

**Figure 6 ijms-21-05831-f006:**
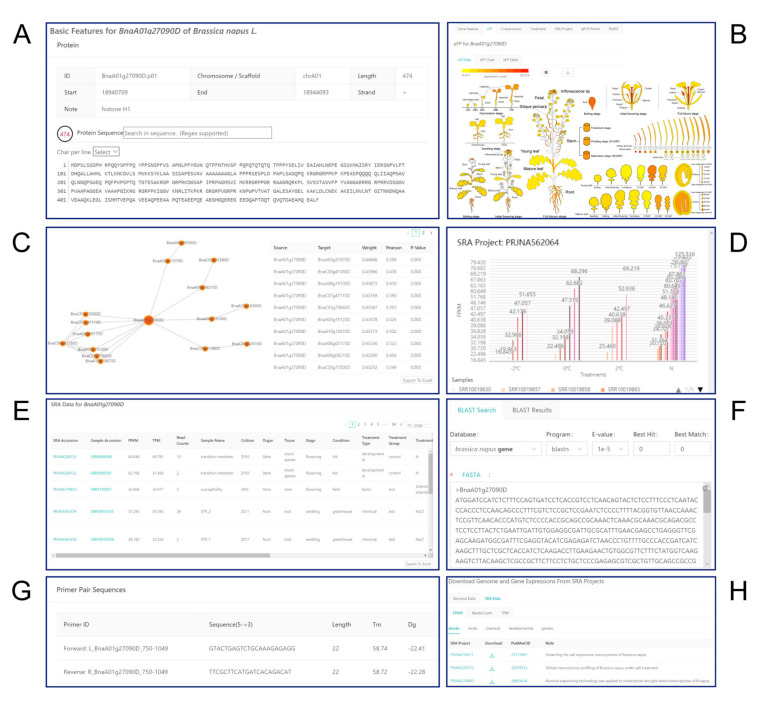
The main functional modules in the BrassicaEDB. (**A**) Gene Feature, (**B**) eFP, (**C**) Coexpression, (**D**) Treatment, (**E**) SRA Project, (**F**) BLAST, (**G**) qPCR primer, (**H**) Downloads.

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
