# Peer review of "BrassicaEDB: A Gene Expression Database for Brassica Crops"

_ijms, 2020, doi:10.3390/ijms21165831_

Round 1
Reviewer 1 Report
Chao et al. report an extensive transcriptomic analysis of Brassica napus development. Their RNA-seq results were combined with public transcriptomic datasets to generate BrassicaEDB, a web-based resource for exploring B. napus transcriptomes.
This is a clear, well-written manuscript, and BrassicaEDB is a valuable resource for plant researchers.
There is however a couple of issues that needs to be addressed; I recommend the authors:
1. To improve the introduction and discussion by including recent studies relevant for the topic, for instance:
- Cheng et al. (Curr Opin Plant Biol. 2017 Apr;36:79-87. doi: 10.1016/j.pbi.2017.01.006) summarized main databases with genomic resources for Brassica;
- Genomic and transcriptomic diversity of B. napus accessions were investigated (Hurgobin et al., Plant Biotechnol J. 2018 Jul;16(7):1265-1274. doi: 10.1111/pbi.12867 ; An et al., Nat Commun. 2019 Jun 28;10(1):2878. doi: 10.1038/s41467-019-10757-1 ), and further used to characterize disease resistance genes (Dolatabadian et al., Plant Biotechnol J. 2020 Apr;18(4):969-982. doi: 10.1111/pbi.13262).
2. To revise the BrassicaEDB contents, since several pages are not working, e.g. <https://biodb.swu.edu.cn/brassica/help>,<https://biodb.swu.edu.cn/brassica/pipeline>.
3. To make available at BrassicaEDB copies of (or stable links to) the reference genome and annotated gene files used in the study pipeline.
4. In the "Downloads" page multiple SRA projects are listed but no experimental details are shown; this complicates identification by researchers of the most useful file for their studies. To facilitate SRA project identification, I suggest to add additional details for each SRA projects, for example by including contents of Table S2.
Minor issues:
- Fig. 4B, replace "Ailgn Reads" with "Align reads" or "Read alignment";
- L. 283, please indicate the Salmon version and if the decoy function was used;
- L. 357, replace the Salmon reference with "Patro et al, Nat Methods. 2017 Apr;14(4):417-419. doi: 10.1038/nmeth.4197."
- It seems to me misleading to state in the title "database for Brassica crops" when in fact only Brassica napus data are included. Should the title be modified?
Reviewer 2 Report
The MS described the expression database of Brassica. This information is valuable for the Brassica research community. However, the MS is poorly described, especially “Materials and Methods”. This section could not include all the analyzed procedures used in this study. Some part of results could be move to “Materials and Methods”. In addition, give detail information about construction and operation of BrassicaEDB. Please refer to below comment for revision.
Line 31-33. Treatment à treatment, or “Treatment”, Coexpression, Gene Feature also corrected. Please check al through MS.
Fig 1 and Fig 1 legend. if you compare between public SRA data and Plant material, the plant material should be changed to other terminology, such in house data?? Please correct it.
Line 89-98 and Fig 2 legend. Please describe it more clearly. Think about move to material and methods.
Line 259-260. Please describe detail information of the seedling, and mature plant. Describe detail information about the location of field.
Reviewer 3 Report
The manuscript presents an interesting and useful new online resource for functional genomics in Brassica species. The gene expression database is conceived to become a very useful tool for the community researcher working with Brassica crops, even if it, at the moment, seems lacking of a prompt query response (I personally looked for the BnaA05g24650D gene but I had no response).
As regards to the manuscript, it is actually about rapeseed crop only and no data are presented for the other 5 Brassica crops included in the datbase, so this aspect should be more clearly stated at the start of the manuscript. Moreover, similar resources are already used by a plenty of researchers for other plant species (e.g. Arabidopsis eFP Browser http://bar.utoronto.ca/efp/cgi-bin/efpWeb.cgi), and these earlier projects should be included in the Background section since they represent a valuable milestone for this research field. Regarding to the methodology used, my biggest concern is that the authors used only 2 biological replicates for the RNAseq study, while usually the minimum number is 3 to provide the acceptable statistical soundness. The use of such a low amount of replicates could not provide the requested robustness for a reference tool as it is claimed by the authors. Furthermore, in the PCA (figure S1) only one replicate is presented for each sample, so it is impossible to appreciate their relative position within the samples and possible mixing among samples. This aspect really hampers a total exploitation of the results which include a very high number of tested conditions across rapeseed plant cycle. Therefore I suggest the authors to re-evaluate the reliability of the produced dataset and evaluate the possibility to publish separately the Brassica EDB database from the RNAseq study.
Round 2
Reviewer 3 Report
Please use "RNA-Seq" term consistently throughout all the manuscript avoiding other similar terms as RNA-seq (caption in Figure 2). Besides, please improve the quality of Figure S4.
Author Response
Please use "RNA-Seq" term consistently throughout all the manuscript avoiding other similar terms as RNA-seq (caption in Figure 2). Besides, please improve the quality of Figure S4.
Respond
Thanks for your comments. We have checked "RNA-Seq" throughout all the manuscript and changed “RNA-seq” to “RNA-Seq” at line 120 and 331, and improved the quality of Figure S4.